# Attitudes toward End-of-Life Resuscitation: A Psychometric Evaluation of a Novel Attitude Scale

**DOI:** 10.3390/healthcare11192618

**Published:** 2023-09-25

**Authors:** Aih-Fung Chiu, Chin-Hua Huang, Chun-Fung Chiu, Chun-Man Hsieh

**Affiliations:** 1Department of Nursing, Meiho University, Pingtung 91202, Taiwan; x00003119@meiho.edu.tw (A.-F.C.); x00003078@meiho.edu.tw (C.-H.H.); 2Kaohsiung Veterans General Hospital, Kaohsiung 81362, Taiwan; cfchiu@vghks.gov.tw; 3Department of Nursing, Tajen University, Pingtung 907101, Taiwan

**Keywords:** attitude, cardiopulmonary resuscitation (CPR), do not resuscitate (DNR), exploratory factor analysis, older adult

## Abstract

Aim. With the advent of an aging society and the development of end-of-life care, there is an increasing need to understand the older generation’s attitude toward end-of-life resuscitation. The study aimed to develop and validate a novel attitude scale toward end-of-life resuscitation in older inpatients. Method. Instrumental development and a psychometric evaluation were used. First, a new attitude scale toward end-of-life resuscitation was formulated from literature views, expert content validity, and face validity. Next, the new scale was evaluated using a principal component analysis and internal consistency reliability in a sample from 106 medical–surgical inpatients in a southern Taiwan hospital 1 enrolled through convenience sampling. Serving as an indicator of concurrent validity, a logistic regression analysis was performed to analyze the association between scores on the scale and intention to discuss end-of-life CPR issues. Results: After being validated by the expert content validity and face validity, a draft of a 20-item scale was created. Throughout the exploratory factor analysis, two items with low factor loadings were removed from the draft scale and an 18-item scale of attitude was generated. This 18-item scale had a three-factor structure that accounted for 64.1% of the total variance; the three components were named ‘stress, avoidance, and ignorance’, ‘a peaceful death’, and ‘self-determination and ambivalence’. The Cronbach’s alpha of the total scale and three components were 0.845, 0.885, 0.879, and 0.857, respectively, which indicated a favorable reliability. Scores on the scale were significantly associated with the intention to discuss end-of-life CPR issues, which also indicated a favorable concurrent validity. Conclusions: A 18-item attitude scale with three factors is a valid scale to measure the attitude toward end-of-life resuscitation. The result provides preliminary evidence of the psychometric properties of the scale. Further research with larger samples or other populations is required.

## 1. Introduction

According to statistics from the Ministry of the Interior, Taiwan’s population of older people over the age of 65 years old exceeded 7% in 1993, making Taiwan close to an ‘aged society’. Then, in 2018, the number of people over 65 years old quickly exceeded 14% and Taiwan officially entered an ‘aging society’. Due to aging and diseases caused by aging, older people may be prone to increasing threats of death. When death comes, different life-support treatment technologies, such as cardiopulmonary resuscitation (CPR), can be used to maintain the heartbeat and prolong vital signs. CPR order is the administration of chest compressions, which are typically performed along with artificial respiration, cardiac defibrillation, or intravenous drug administration. The primary purpose of CPR is to save or sustain life. During an event of cardiac or respiratory arrest, CPR is typically required [1,2]. However, CPR for in-hospital cardiac arrest has been reported to have a low 1-year survival rate [3]. For older people approaching the end of life, CPR lacks curative efficacy and may cause pain and suffering for patients. In addition, unnecessary medical treatment not only imposes psychological and financial burdens on families but also creates ethical dilemmas for healthcare providers involved in the end-of-life decisions of patients. Controversially, the do-not-resuscitate (DNR) order, also known as the ‘no code’ or ‘allow natural death order’, is an order to not perform CPR in the event of a cardiac or respiratory arrest. The DNR order was initially introduced to medical practice to prevent suffering and ensure the autonomy of patients at the end of their life, decrease the families’ dilemma when they make decisions for their loved ones, and prevent medical futility [1,4]. Despite this, a DNR remains a delicate and contentious issue that puts healthcare providers in morally ambiguous situations.

In recent decades, legal circles, medical circles, and nongovernmental organizations have been continuously committed to the promotion of laws or regulations related to end-of-life care. For example, the ‘Hospice Palliative Care Act’ of 2000 protects the will of terminally ill patients regarding their medical treatments, which specifically stipulates healthcare providers to respect terminally ill patients’ will on the medical treatment. In 2015, the ‘Patient Autonomy Act’, the first legislation about the right of a good death in Asia, was then enacted, granting people the autonomy to elect to die with dignity. Though end-of-life care is being increasingly discussed, talking about death is still taboo. The rate of making a medical decision or signing an end-of-life DNR order in advance is varied. For example, an article conducted a review of 150 studies with 795,909 participants in the United States between 2011 and 2016 and found that about 38.2% of chronic disease patients and 32.7% of healthy adults completed advanced medical directives [4]. Another study collected data from 1136 patients through the electronic medical records of 13 hospitals and found that only 27.5% of patients documented their DNR directives, while 63.7% were full code, and 8.8% had no directive [5]. In Taiwan, a study of 206 medical and surgical inpatients over the age of 20 reported that 52.0% of the respondents would sign their advance medical directive in the future, 15.5% would not sign it, and 32.5% of participants were undecided [6]. Recently, a study analyzing data from 1338 terminal cancer patients reported that 56.35% had their DNR documents [7].

Attitudes are the positive and negative evaluations that individuals hold about specific behaviors. A person’s attitudes can affect how individuals perceive an object and reflect psychological tendencies, which affect their behavior [7,8,9]. In other words, when people’s attitude towards end-of-life CPR/DNR orders becomes more positive, their behavioral intent to discuss or sign a DNR order or other advance directives increases [9]. Conversely, if people have a negative attitude toward end-of-life CPR/DNR orders, the preferential behavioral intention decreases accordingly. Patients’ attitudes toward end-of-life CPR/DNR orders are widely varied. When death is imminent, people strongly prefer decisions that ensure comfort or alleviate suffering, rather than invasive treatments to prolong life. A peaceful and pain-free death with dignity is considered a basic requirement of end-of-life care [8,10]. However, there are still many negative attitudes and struggles toward it. For example, some people considered that discussions about DNR orders were stressful [8]. They feared that they would be undertreated or neglected if they signed a DNR order and were also worried about the possible abuse or wrongful interpretation of the DNR order by physicians and relatives [11]. Indeed, empirical evidence showed that patients hospitalized with acute heart failure and with DNR orders were significantly less likely to have received any quality treatment than those without DNR orders [12]. Other people might not want to talk about end-of-life DNR orders because they consider it unnecessary, or too early, even in terminal cancer patients [7,13]. Instead, they prefer to designate a family member to act as a surrogate decision-maker [14], which is particularly prevalent in Asian societies. However, there are always differences in their opinions regarding end-of-life issues between patients and their families [15]. Ambivalence regarding preference for or refusal of end-of-life resuscitation also exists. Some people believe that cardiopulmonary resuscitation has a certain chance of successful resuscitation in older cancer patients [16]. On the contrary, some people might worry that their death would cause the emotional distress of their loved ones [17,18,19]. Thus, they might show low autonomy in making end-of-life APR/DNR orders [20].

To our knowledge, previous studies measuring the attitude toward end-of-life CPR/DNR orders were usually focused on professional views [21,22,23]. Few studies have been undertaken on older inpatients’ views in Taiwan, and no appropriate attitude scale has been identified. A tool to measure inpatients’ attitudes toward end-of-life CPR/DNR orders is crucial to ensuring positive end-of-life care outcomes. Thus, the present study was conducted to developing a Mandarin Chinese version of an instrument that can provide information regarding attitudes toward end-of-life CPR/DNR orders among older inpatients of a hospital in southern Taiwan.

## 2. Aim

The aim of the present study was to develop a novel instrument of a Mandarin Chinese version of attitudes scale regarding end-of-life CPR/DNR order and examining its psychometric property using a convenient sample of older inpatients of a hospital in southern Taiwan.

## 3. Methods

Two steps for the development of the novel instrument were conducted, including the developmental stage and the psychometric validation which was conducted with exploratory factor analysis (EFA), internal reliability testing (Cronbach’s alpha), and the concurrent validity. One-sample *t*-tests with a set point of 4 were used to detect the levels of attitude scores.

### 3.1. The Developmental Stage

Because the previous instruments found to be available were not appropriately meeting the purposes of our study, we created our own. Examples included measures such as the Preferences for Care near the End of Life scale, which had too many dimensions and was devised with items from university students and community-residing adults instead of older adults [24]. Thus, we aimed to develop a self-administrative tool for our needs. In order to create the initial item pool regarding patients’ attitudes toward end-of-life CPR/DNR orders, researchers searched relevant papers and measures using the keywords ‘attitude’, ‘end of life’, ‘resuscitation’, ‘cardiopulmonary resuscitation’, or ‘do not resuscitation’ in the title or abstract of the databases of MEDLINE, CINAHL, and google scholar search. Researchers first identified whether items fit the construct we wanted to survey. In addition, 10 inpatients were interviewed and asked how they felt about the death and the end-of-life CPR/DNR orders. All initial items were created and examined carefully by researchers. If any items did not fit the topic of our study, had overlapping content, or expressed similarities, they would be deleted, edited, or revised to ensure clarity and relevance. The remaining 22 items were then further tested by content validity.

Content validity index (CVI), the most widely reported approach for content validity in instrument development, refers to the degree to which an assessment instrument is relevant to, and representative of, the targeted construct it is designed to measure. Using the recommendations by Lynn (1986) [25], a panel of four experts in health care, elderly care, and spiritual care was invited to assess whether the 22-item scale was relevant to the construct we wanted to measure. Then, experts rated each item using a 4-point scale (1 = not relevant, 2 = somewhat relevant, but this item needs to be majorly revised or deleted; 3 = relevant but needs minor revision, and 4 = very relevant). CVI can be computed using the item-level -CVI (I-CVI) and the scale-level CVI (S-CVI). I-CVI is computed as the number of experts giving a score of 3 or 4 for each item divided by the total number of experts. The items with a CVI score of <0.75 were removed (2 out of 4 experts scored 1 or 2), being deemed too sensitive or vague. Similarly, S-CVI is calculated using the number of items in a tool that have achieved a score of 3 or 4. The final S-CVI value was 1.00, reaching the standard of 0.8 [25]. Thus, the expert panel agreed on a draft of a 20-item attitude scale.

The 20-item attitude scale included positive and negative statements, in which items were scored on a 5-point Likert scale. In positively worded statements, the response of ‘strongly disagree’ to ‘strongly agree’ were given 1 to 5 points, respectively. The negative worded statements were calculated with reverse scoring. The higher the score, the more positive the attitude. On the other hand, for assessing participants’ characteristics for our study sample, social-demographic data were collected, including age, gender, medical–surgical diagnosis, education level, presence of chronic diseases, and the occurrence of a recent death of a family member or close friend. Three other items were also collected (i.e., ‘Have you heard of CPR/DNR orders’ for information collection, ‘Have you the intention to discuss your end-of-life CPR/DNR decisions’ for preferences regarding CPR/DNR conversations and ‘Have you ever signed a DNR order or other advance decision form’ for the signing ratio of DNR) and were binary-response items (yes or no responses).

### 3.2. The Psychometric Validation

After the initial attitude scale was created, a survey with a cross-sectional design and convenience samples was administered between September and December 2019. The inclusion criteria were inpatients who were aged 60 years or older and could complete the questionnaire in the medical or surgical wards of a teaching hospital in southern Taiwan. After explaining the purpose of our study and getting the written informed consent, we handed the questionnaire to participants to fill out. If the participants were unable to fill it out by themselves, the research assistant would read the items one by one and ask them to choose the option that best fits their perception. Those who could not provide informed consent or could not complete the questionnaire were excluded. In total, 106 inpatients were enrolled and completed the 20-item attitude scale. Based on the recommendations of Kline, the sample size required for EFA was two to five participants per item [26]; our study sample met this requirement.

A series of the EFA was used to test the construct validity. With a Kaiser–Meyer–Olkin index score of 0.78 and a significant result for Bartlett’s test of sphericity (χ^2^ = 1327.36, *p* < 0.001) in the 20-item scale, it indicated that the data were appropriate for analysis. Using the principal component analysis, varimax rotation, and Cattell’s scree plot, factors were extracted and named by theme [27]. In the meanwhile, Cronbach’s alpha values were used to determine the internal consistency reliability. A Cronbach’s alpha of ≥0.8 was considered to indicate consistency [28]. In addition, one-sample *t*-tests with a set point of 4 were used to understand whether the mean scores of attitude scale and three domains reached a certain standard.

As an indicator of concurrent validity, the binary-response item ‘Have you the intention to discuss your end-of-life CPR/DNR decisions?’ (yes or no response) was associated with the scores of this novel scale using univariate and multiple logistic analyses. In addition, a one-sample *t*-test was used to determine whether the mean of participants’ attitudes was different from 4 points (the level of agreement on a 5-point Likert scale). All collected data were analyzed using IBM SPSS version 23.0.

### 3.3. Ethical Considerations

This study was approved by the Human Research Ethics Committee of Kaohsiung Veterans General Hospital, and written informed consent was obtained from all participants beforehand.

## 4. Results

### 4.1. Participant Characteristics

A total of 106 participants (56 women and 50 men), with a mean age of 74.5 ± 10.1 (range: 60–98) years, responded to the quantitative survey. Table 1 revealed that nearly half of the participants (49.1%) were enrolled from medical wards and 61.3% had low education attainment of ≤6 years. More than half (57.5%) had a ≥1 score of CCI and half (50.9%) participated in medical decision-making during hospitalization. About one-third (33.0%) had experienced the death of a family member or close friend in recent years. A total of 64.2% reported that they had ever heard about the CPR/DNR orders, and 61.3% indicated that they had the intention to discuss their end-of-life CPR/DNR decisions, while only 10% of them had already signed a DNR order or advance decision forms.

### 4.2. EFA and Reliability Analysis

Table 2 presents the results of the EFA and reliability analysis. Two items, ‘One’s life should not be meaninglessly prolonged’ and ‘At the end of life, CPR can increase my chances of survival’, had a low factor loading, and they were removed from the 20-item draft of the attitude scale, resulting in a final 18-item attitude scale (Table 2). With a Kaiser–Meyer–Olkin index score of 0.77 and a significant result for Bartlett’s test of sphericity (χ^2^ = 1242.75, *p* < 0.001) in the 18-item scale, three factors were extracted and named ‘stress, avoidance or ignorance’ (eight items), peaceful death (six items), and ‘self-determination and ambivalence’ (four items). The factor loadings of all items were >0.6, which is higher than the recommended threshold of 0.4 and indicates a close relationship between the factors and items [27]. These three factors explained 62.92% of the total observed variance. The reliability analysis revealed that the Cronbach’s alpha values were 0.845 for the overall scale, 0.885 for factor 1 (stress, avoidance, and ignorance), 0.879 for factor 2 (peaceful death), and 0.857 for factor 3 (self-determination and ambivalence), indicating a highly favorable internal consistency [28].

### 4.3. The Scores of the Attitude toward End-of-Life Resuscitation

Table 3 shows the average scores of the attitude toward end-of-life CPR/DNR orders. To compare the three factors, reverse scoring was employed to calculate factor 1 items, which were associated with negative perceptions of a CPR/DNR order at the end of life. The average scores of the overall scale and its three factors were 3.66 (standard deviation [SD] = 0.50), 3.29 (0.81), 4.14 (0.64), and 3.68 (0.74), respectively. Using a one-sample *t*-test with a set point of four, the scores of the total scale, factor 1, and factor 3 had significantly lower scores than we expected, implying the total attitude and attitude in both domains were not optimal. On the other hand, factor 2 had significantly higher scores.

### 4.4. Association of the Total Attitude Scale and Three Factors’ Scores with the Intention to Discuss the End-of-Life CPR/DNR Orders

Table 4 presents the results from the univariate and multiple logistic regression models. In the univariate logistic regression model, the overall attitude score and factors 1 and 3 were significantly associated with the intention to discuss end-of-life CPR/DNR orders; the odd ratios (95% confidence intervals, CI) were 5.08 (1.91–13.49), 2.25 (1.29–3.94), and 1.93 (1.09–3.41), respectively, indicating that the attitude scale had a favorable concurrent validity. After adjusting for all three factors, only factor 1 ‘stress, avoidance and ignorance’ was observed to be significantly associated with the intention to discuss end-of-life CPR/DNR orders; the adjust odds ratio (95% CI) was 2.22 (1.26–3.93).

## 5. Discussion

In this study, patients over 60 years of age who were inpatients in medical and surgical departments were enrolled to develop an attitude scale regarding CPR/DNR at the end of life. This 18-item Mandarin Chinese attitude scale included 8 items for ‘stress, avoidance or ignorance’, 6 items for ‘peaceful death’, and 4 items for ‘self-determination and ambivalence’. The strength of the scale came from the fact that its items were generated based on a literature review, patients’ views, expert content validity testing, construct validation testing (based on EFA), internal consistency reliability testing (based on Cronbach’s alpha) (Table 2), and concurrent validity testing (Table 4), suggesting that this scale and its three factors exhibited favorable reliability and validity. Our results supported the usefulness of this novel scale of attitude regarding end-of-life CPR/DNR orders among older inpatients.

The following is a discussion of the narrative in the text of the attitude scale (Table 3). First, the factor of ‘stress, avoidance, and ignorance’ had a significantly lower mean score and contributed the most explained variation of this attitude scale (Table 2), implying that our study samples had a lot of negative perceptions regarding end-of-life CPR/DNR orders, e.g., the feelings of being distressed and avoidant thinking and behaviors. The results are congruent with previous studies, in which people considered that discussions about DNR orders were stressful, being worried about the possible undertreatment, neglect, or even abuse from the process of treatment [8,11,12]. Sometimes, in order to relieve the distress and uneasiness, this stress would be modulated by defense mechanisms (e.g., inappropriate thinking or behavior, thinking that it is too early or unnecessary to make an end-of-life decision, or belief that it is others’ responsibility) [7,8,13,14,17]. In addition, after putting all three factors into a logistical regression with the enter method, the factor of ‘stress, avoidance, and ignorance’ was the only independent predictor of participating in the discussion regarding end-of-life CPR/DNR orders (Table 4). Namely, negative attitudes toward ‘stress, avoidance, and ignorance’ were the major factor in impeding participation in discussions regarding end-of-life CPR/DNR orders. Thus, healthcare providers engaged in end-of-life care should be aware of these barriers and create strategies to reduce barriers regarding end-of-life CPR/DNR that need to be figured out.

In terms of the factor of ‘peaceful death’, the result showed a significantly higher score in the factor, that is, most people wanted to die peacefully, pain-free, naturally, and without spiritual suffering, especially in the oldest old people [17]. A review article introduced and demonstrated the concept of ‘good death’, in which the major themes included a favorite dying process (94% of reports), pain-free status (81%), emotional well-being (64%), and so on [29]. Namely, our results in the factor ’peaceful death’ met the core element of a good death, though this factor was not associated with the intention of participating in discussions regarding end-of-life CPR/DNR orders shown in Table 2.

Lastly, the third factor ‘self-determination and ambivalence’ was extracted and contributed 9.85% of the explained variance. The items included the statements of discomfort toward discussions regarding CPR/DNR decisions with doctors or families, and the conflicting emotions caused by death. Previous studies have stressed that the ability to control their death (e.g., making decisions and plans) was crucial when discussing end-of-life care [21,29,30]. Because the medical conversation provided by healthcare professionals could contribute to the transition process from curative to palliative care [31], it is necessary to strengthen and proactively initiate communication with elderly patients in general medical and surgical departments to eliminate their doubts about CPR/DNR orders, to facilitate the promotion of end-of-life care and patients’ autonomy.

In the present study, only 64.2% of participants had ever heard of DNR orders and 61.0% had the intention to discuss their end-of-life DNR decisions. The results were higher than the report of a previous study conducted in Saudi Arabia, which reported that 49.6% of participants had heard of a DNR before [8]. Indeed, in the past 10 years, with the progress of Taiwan’s medical education and the establishment of related laws regarding advance directives, Taiwanese people have become increasingly aware of the importance of end-of-life care and patient autonomy. The number of Taiwanese signing advanced medical directives is increasing year by year [32]. However, only 9.4% of our study samples had signed a DNR order (Table 1). Previous studies demonstrated that medical condition is one of the most important factors affecting the signature of a DNR order [7,8,13,17]. The low percentage can be explained by the reason that our study sample came from general medical and surgical inpatients who are not at immediate risk of death. In addition, according to the knowledge–attitude–behavior theory, which demonstrated that an individual’s attitude would affect their exhibited behavior, our study samples with poor attitudes might contribute to the low percentage of signing the DNR orders. Thus, further efforts to raise awareness about the importance of CPR/DNR orders and promote better attitudes are needed.

## 6. Future Lines of Research and Practical Applications

While many studies measuring attitudes toward end-of-life CPR/DNR orders have focused on professional healthcare providers [21,22,23], this study provides preliminary evidence supporting the use of straightforward measures to assess attitudes toward this issue from the patient’s view. This novel scale can help healthcare providers examine and recognize older patients’ attitudes toward end-of-life CPR/DNR orders; it can also help them initiate and engage with patients in a respectful manner to discuss their end-of-life CPR/DNR order, which can help improve their end-of-life care and ensure patients’ autonomy. This attitude scale can be used in large-scale surveys to elicit awareness toward end-of-life CPR/DNR orders among older people. Moreover, researchers can utilize this tool to assess the effects of interventions on the attitudes of patients or to explore differences in attitudes between populations.

## 7. Limitations of the Study

Some limitations must be considered when the results are interpreted. Firstly, the samples are potentially limited on a convenience basis. We only collected inpatients from the medical and surgical wards of a certain hospital who were willing to provide informed consent. We did not consider patients who were undergoing major surgery, pain, or those who had psychiatric conditions such as dementia, which may influence the opinions expressed. To increase the generalizability and applicability of studies involving these subjects, more diverse and larger samples may be needed. Second, we performed EFA to determine the structure of attitudes toward end-of-life CPR/DNR orders. The scale should be further tested through a confirmatory factor analysis on a different sample; additionally, the scale’s correlation with other similar scales and its test–retest reproducibility (i.e., the temporal stability of an instrument over time), which are also considered to be important indicators of validity, could be studied further. Third, this scale focuses on the attitude regarding end-of-life CPR/DNR orders but does not account for all possible facets of treatment directives, and further studies assessing other end-of-life issues may be needed. Finally, because of the cross-sectional design of this study, the causality among the examined variables could not be established.

## 8. Conclusions

This study provided psychometric evidence supporting the validity and reliability of a scale for assessing attitudes toward end-of-life CPR/DNR orders. The scale comprises three dimensions pertaining to the views of inpatients; it can be used to assess and understand the attitudes of inpatients toward end-of-life CPR/DNR orders and thereby ensures patients’ autonomy and improves end-of-life care. The low mean scores on the attitude scale, especially in both factors of ‘stress, avoidance, and ignorance’, and ‘self-determination and ambivalence’, suggest that our study samples’ attitude was negative, and effort is required to promote end-of-life care.

## Figures and Tables

**Table 1 healthcare-11-02618-t001:** Demographics of participants (*n* = 106).

Variable	Mean	±SD
Age (years, mean ± standard deviation)	74.5	±10.1
60–<75 (*n*, %)	63	(59.4)
≥75	43	(40.6)
Sex (*n*, %)		
Female	56	(52.8)
Male	50	(47.2)
Inpatient department (*n*, %)		
Medicine	52	(49.1)
Surgery	54	(50.9)
Education attainment (*n*, %)		
≤6 years	65	(61.3)
>6 years	41	(38.7)
Charlson Comorbidity Index (*n*, %)		
<1	45	(42.5)
≥1	61	(57.5)
Participate in medical decision-making during hospitalization (*n*, %)		
No	52	(49.1)
Yes	54	(50.9)
Death of a family member or close friend in a recent year (*n*, %)		
No	71	(67.0)
Yes	35	(33.0)
Heard of CPR/DNR order (*n*, %)		
No	38	(35.8)
Yes	68	(64.2)
Intention to discuss the end-of-life CPR/DNR decisions (*n*, %)		
No	41	(38.7)
Yes	65	(61.3)
Signing of an DNR order or other advance decision form		
No	96	(90.6)
Yes	10	(9.4)

**Table 2 healthcare-11-02618-t002:** Standardized factor loadings, explained variance (obtained using the exploratory factor analysis), and Cronbach’s alpha of scale (*n* = 106).

	Mean	±SD	Factor 1	Factor 2	Factor 3
Factor 1: Stress, avoidance, or ignorance ^a^					
1. I feel stressed when I discuss CPR or intubation at the end of life.	3.26	±1.12	0.841		
2. It is far too early to discuss whether I should undergo CPR or intubation at the end of my life.	3.30	±1.06	0.780		
3. At the end of my life, it will be better to let others decide whether I receive CPR or intubation.	3.63	±1.06	0.763		
4. Whether to receive CPR or intubation at the end of my life may bring me misfortune.	3.27	±1.16	0.749		
5. At the end of life, deciding whether a patient undergoes CPR is the responsibility of their family members.	3.46	±1.07	0.720		
6. If someone signs a DNR order, they can revoke the decision at the end of their life.	3.13	±1.16	0.718		
7. If someone signs a DNR order, physicians will not treat the individual when he/she is severely ill.	3.19	±1.10	0.711		
8. Even if I don’t make the decision of end-of-life CPR/DNR, there will be a way when the time comes	3.03	1.02	0.644		
Factor 2: Peaceful death					
9. At the end of life, a DNR order would allow me to pass away peacefully and with dignity.	4.16	±0.82		0.814	
10. At the end of life, CPR is spiritual torture.	4.20	±0.81		0.805	
11. At the end of life, a person must die painlessly with their body intact	4.15	±0.75		0.791	
12. At the end of life, CPR merely prolongs the dying process and has no meaningful effect.	4.02	±0.87		0.761	
13. At the end of life, CPR worsens pain.	4.22	±0.77		0.752	
14. At the end of life, I will let destiny run its course.	4.07	±0.85		0.701	
Factor 3: Self-determination and ambivalence					
15. I can comfortably discuss whether I should undergo CPR at the end of my life with my family.	3.55	±0.98			0.872
16. I can comfortably discuss whether I should undergo CPR at the end of my life with my doctor.	3.58	±0.94			0.855
17. If I make end-of-life decisions in advance, I will not be a burden to others	3.84	±0.79			0.793
18. If I have a DNR order and am intubated or resuscitated, physicians can extubate or stop CPR.	3.75	±0.79			0.630
Explained variance (%)			28.57	24.50	9.85
Cronbach’s alpha (total Cronbach’s alpha: 0.845)			0.885	0.879	0.857

^a^ Reverse coded for analysis.

**Table 3 healthcare-11-02618-t003:** Average scores of overall attitude scale and each factor (*n* = 106).

	Number of Items	Mean	SD	*p*-Value ^a^
Average of overall attitude score	18	3.66	±0.50	<0.001 *
Factor 1: Stress, avoidance, or ignorance ^a^	8	3.29	±0.81	<0.001 *
Factor 2: Peaceful death	6	4.14	±0.64	0.032 *
Factor 3: Self-determination and ambivalence	4	3.68	±0.74	<0.001 *

* *p* < 0.05. One-sample *t*-tests were used. ^a^ Reverse coded for analysis.

**Table 4 healthcare-11-02618-t004:** Association of total attitude scale and three factors’ scores with the intention to discuss the end-of-life CPR/DNR orders (*n* = 106).

	Univariate Analysis			Multiple Analysis		
Variables	OR	(95% CI)	*p*-Value	AOR	(95% CI)	*p*-Value
Overall attitude score	5.08	(1.91–13.49)	0.001 *			
Factor 1: Stress, avoidance, and ignorance ^a^	2.25	(1.29–3.94)	0.004 *	2.22	(1.26–3.93)	0.006 *
Factor 2: Peaceful death	1.48	(0.80–2.75)	0.210	1.28	(0.62–2.64)	0.500
Factor 3: Self-determination and ambivalence	1.93	(1.09–3.41)	0.023 *	1.64	(0.87–3.11)	0.129

* *p* < 0.05. CI, confidence interval. OR, odds ratio. AOR, adjust odds ratio for all variables included in the multiple analysis. ^a^ Reverse coded for analysis

## Data Availability

Data are available upon request.

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
