# Peer review of "Attitudes toward End-of-Life Resuscitation: A Psychometric Evaluation of a Novel Attitude Scale"

_healthcare, 2023, doi:10.3390/healthcare11192618_

Round 1
Reviewer 1 Report
Dear Authors,
thank you very much for the opportunity to review the article entitled Attitudes toward end-of-life resuscitation: a psychometric evaluation of a novel attitude scale. It raises extremely important issues and aims to provide psychometric evidence supporting the validity and reliability of the scale for assessing attitudes toward end-of-life CPR/DNR orders what is highly important matter. The text of the manuscript itself is also written in an understandable and clear way.
I my opinion the study was quite well planned and the presented data and analyzes were performed reliably. After a thorough analysis and review process, I would like to ask for clarification on a few important points:
1. In lines 155-161 it is mentioned that 100 patients aged 65 years or older from the medical or surgical wards were enrolled in the study. How were these patients selected? Was it a hundred consecutive patients, who did not meet the exclusion criteria, admitted to the indicated departments? Please kindly clarify this important issue.
2. Lines 158-159 mention the criteria for exclusion from the study: "Those who could not complete the questionnaire because of delirium, terminal medical conditions, or other problems were excluded." Was the technical inability to complete the questionnaire the only exclusion criterion? What does "or other problems" mean? Has the patient's current mental state been taken into account? Have any chronic psychiatric conditions been considered? Fear of major surgery, pain, psychiatric disorders or dementia may influence the opinions expressed in the survey. I kindly ask you to explain and specify these issues in the methodology or in the paragraph - limitations of the study.
3. Chapter introduction and discussion may also require some care. In the opinion of the reviewers, a better selection of references will improve them. Only 8 of the 36 papers cited in the peer-reviewed article have been published in the last five years. Thus, the introduction and discussion could gain in quality after a more thorough review of the literature and reference to the latest publications on the subject. Perhaps increasing the number of searched keywords will make it easier to find at least a few scientific papers from recent years that are related to the attitude toward end-of-life CPR/DNR orders? Also in older patients' population. Also in the population of elderly patients or at least close to it or at least including patients over 65 years of age. Finding and referring to such recent scientific papers could improve the quality of the introduction and discussion.
A minor review and suggestions:
- verses e.g. 44, 74, 85, 97, 200 - no spaces before brackets with citation number
- line 110 - a different font / size was used
- verses 76 and 149 - most likely erroneous highlighting of parts of the text
Thank you again for the opportunity to review an article on this very important issues. I sincerely hope that this review will allow you to draw constructive conclusions that will prove useful.
Author Response
Response to Reviewer 1
Thank you very much for the opportunity to review the article entitled Attitudes toward end-of-life resuscitation: a psychometric evaluation of a novel attitude scale. It raises extremely important issues and aims to provide psychometric evidence supporting the validity and reliability of the scale for assessing attitudes toward end-of-life CPR/DNR orders what is highly important matter. The text of the manuscript itself is also written in an understandable and clear way.
I my opinion the study was quite well planned and the presented data and analyzes were performed reliably. After a thorough analysis and review process, I would like to ask for clarification on a few important points:
- In lines 155-161 it is mentioned that 100 patients aged 65 years or older from the medical or surgical wards were enrolled in the study. How were these patients selected? Was it a hundred consecutive patients, who did not meet the exclusion criteria, admitted to the indicated departments? Please kindly clarify this important issue.
Reply: Thank you for your comments. We have added the description of the process of enrolling participants. Please see L169-175.
- Lines 158-159 mention the criteria for exclusion from the study: "Those who could not complete the questionnaire because of delirium, terminal medical conditions, or other problems were excluded." Was the technical inability to complete the questionnaire the only exclusion criterion? What does "or other problems" mean? Has the patient's current mental state been taken into account? Have any chronic psychiatric conditions been considered? Fear of major surgery, pain, psychiatric disorders or dementia may influence the opinions expressed in the survey. I kindly ask you to explain and specify these issues in the methodology or in the paragraph - limitations of the study.
Reply: Thank you for your comments. We have deleted some inappropriate descriptions, rewritten the process of enrolling participants, and added some research limitation. Please see L169-175 and L331-334.
- Chapter introduction and discussion may also require some care. In the opinion of the reviewers, a better selection of references will improve them. Only 8 of the 36 papers cited in the peer-reviewed article have been published in the last five years. Thus, the introduction and discussion could gain in quality after a more thorough review of the literature and reference to the latest publications on the subject. Perhaps increasing the number of searched keywords will make it easier to find at least a few scientific papers from recent years that are related to the attitude toward end-of-life CPR/DNR orders? Also in older patients' population. Also in the population of elderly patients or at least close to it or at least including patients over 65 years of age. Finding and referring to such recent scientific papers could improve the quality of the introduction and discussion.
Reply: Thank you for your comments. We have reviewed the relevant literature to try to improve the quality of the introduction and discussion. Please see L 89 – 93, L97-104, L 261 – 314.
- A minor review and suggestions:
- verses e.g. 44, 74, 85, 97, 200 - no spaces before brackets with citation number
- line 110 - a different font / size was used
- verses 76 and 149 - most likely erroneous highlighting of parts of the text
Reply: Thank you for your comments. We have revised these errors.

Reviewer 2 Report
Dear authors,
I appreciate the opportunity to review this study. In this case, the authors conduct a psychometric scale for the measurement of attitudes toward a controversial topic.
Introduction: The authors provide information relevant to the study. However, it is striking that so few references on the subject are used, so I consider that it would be appropriate to review this point and expand the bibliography, in order to improve the paper.
Methodology. The methodology proposed is consistent with the objectives and the results obtained. The authors have submitted the scale to the necessary validation processes.
The study itself is interesting because the topic addressed is relevant to the health, emotional and mental health care required by elderly patients.
Discussion: This section lacks an exhaustive discussion with previous studies. Again, there is little literature review work, and in this section it is essential.
It is also recommended that a section on Limitations and another on Future lines of research and practical applications be added.
Author Response
Response to Reviewer 2
Dear authors,
I appreciate the opportunity to review this study. In this case, the authors conduct a psychometric scale for the measurement of attitudes toward a controversial topic.
- Introduction: The authors provide information relevant to the study. However, it is striking that so few references on the subject are used, so I consider that it would be appropriate to review this point and expand the bibliography, in order to improve the paper.
Reply: Thank you for your comments. We have reviewed relevant literature. Please see L82-84, L89-101.
- The methodology proposed is consistent with the objectives and the results obtained. The authors have submitted the scale to the necessary validation processes.
Reply: Thank you for your comments.
- The study itself is interesting because the topic addressed is relevant to the health, emotional and mental health care required by elderly patients.
Reply: Thank you for your comments.
- Discussion: This section lacks an exhaustive discussion with previous studies. Again, there is little literature review work, and in this section it is essential.
Reply: Thank you for your comments. We have reviewed relevant literature. Please see L250-252, L261-273, L274-314.
- It is also recommended that a section on Limitations and another on Future lines of research and practical applications be added.
Reply: Thank you for your comments. We have added the sections of application and limitations. Please see L315 and L327, 331-334.

Reviewer 3 Report
1º Yoy said: "The items with a CVI score of <0.75 were removed or revised". How many and what have these items been? What were the variations to which they were subjected?
2º I understand that the validation work refers to the Mandarin Chinese language, but I have not seen this reflected in the work, which can lead to confusion because, of course, the results have been presented in English.
3º In the summary you say that “The Cronbach’s alpha of the total scale and three components 23 was between 0.834 and 0.904,” but in table 2 of the results the following appear: 0.879; 0.873; 0.857; for the three components and 0.838 for the total scale (what is the reason for this discrepancy in results?).
Positive points are:
1º work directly with vulnerable population
2º Psychometric requirements, for example, “These three factors explained 62.36% of the total observed variance” are highly acceptable.
A very interesting job.
Author Response
Response to Reviewer 3
1º You said: "The items with a CVI score of <0.75 were removed or revised". How many and what have these items been? What were the variations to which they were subjected?
Reply: Thank you for your comments. We have revised our manuscript. Please see L148-149, L211-216.
2º I understand that the validation work refers to the Mandarin Chinese language, but I have not seen this reflected in the work, which can lead to confusion because, of course, the results have been presented in English.
Reply: Thank you for your comments. We have revised our manuscript and stressed the Mandarin Chinese language. Please see L107, L111-112.
3º In the summary you say that “The Cronbach’s alpha of the total scale and three components 23 was between 0.834 and 0.904,” but in table 2 of the results the following appear: 0.879; 0.873; 0.857; for the three components and 0.838 for the total scale (what is the reason for this discrepancy in results?).
Reply: Thank you for your comments. We have corrected errors in our manuscript. Please see L220-224.
Positive points are:
1º work directly with vulnerable population
2º Psychometric requirements, for example, “These three factors explained 62.36% of the total observed variance” are highly acceptable.
A very interesting job.

Round 2
Reviewer 1 Report
Dear Authors,
Thank you very much for the opportunity to review the article entitled "Attitudes toward end-of-life resuscitation: a psychometric evaluation of a novel attitude scale" again. It raises extremely important issues and aims to provide psychometric evidence supporting the validity and reliability of the scale for assessing attitudes toward end-of-life CPR/DNR orders and I my opinion the study was quite well planned and the presented data and analyzes were performed reliably.
I have reviewed the modified manuscript point by point and appreciate all the improvements and clarifications provided by the authors. I also believe that they had a constructive impact on the entire work and its scientific significance. The only comment I would like to add now is to wait for the progress of research by authors on the same topic. Therefore, I encourage you to expand the participating centers, increase the number of study groups and refine the methodology even more. This will allow us to make important progress in this field.